# The Relationship of Balance Disorders with Falling, the Effect of Health Problems, and Social Life on Postural Balance in the Elderly Living in a District in Turkey

**DOI:** 10.3390/geriatrics4020037

**Published:** 2019-05-17

**Authors:** Tahsin Barış Değer, Zeliha Fulden Saraç, Emine Sumru Savaş, Selahattin Fehmi Akçiçek

**Affiliations:** 1Directorate of Health Affairs, Söke Municipality, Söke, Aydın 09200, Turkey; 2Geriatrics Section, Faculty of Medicine, Ege University, Bornova, Izmir 35100, Turkey; fulden.sarac@ege.edu.tr (Z.F.S.); emine.sumru.savas@ege.edu.tr (E.S.S.); fehmi.akcicek@gmail.com (S.F.A.)

**Keywords:** balance disorder, prevalence, elderly, fall, medical conditions, social mobility

## Abstract

The aim of this study was to determine the prevalence of balance disorders; the effects of sociodemographic, medical, and social conditions on postural balance; and the relationship between balance and falls in elderly individuals. The study design was cross-sectional. A total of 607 community-dwelling elderly individuals with a mean age of 73.99 ± 6.6 years were enrolled after being selected by stratified random sampling. The study was performed as a face-to-face survey in the homes of elderly individuals. Sociodemographic and medical data were obtained from elderly individuals using the Elderly Identification Form. Balance disorders were determined using the Berg Balance Scale (BBS). In this study, the prevalence of balance disorders was found to be 34.3% in the community-dwelling elderly. Older age, physical disability, having four or more chronic illnesses, the presence of incontinence, having a history of falls, not walking regularly, absence of free time activity, and obesity were found to be associated with an increased prevalence of balance disorders. Balance disorders are commonly seen in the elderly and may be triggered by a variety of biological and social factors. It is crucial to develop and implement national health and social policies to eliminate the causes of this problem, as well as to prioritize preventive health services in the ever-increasing elderly population.

## 1. Introduction

Balance is the ability to collect sensory and proprioceptive signals related to a person’s position in space and to produce the appropriate motor responses to control body movement [1]. When this ability deteriorates, due to both disease and the normal aging process, the risk of falling increases in the elderly [2]. Balance disorders are one of the most important reasons leading to falls [3]. Falling increases the possibility of death and disability; furthermore, it may cause the loss of independence [4]. In 2014, an estimated 29 million falls were reported in the US. Twenty-seven thousand older adults died, and 7 million were injured because of falls [5]. Approximately 68% of hospitalizations of injured elderly individuals were reported to be because of falls, and this rate reached 86% in individuals aged ≥85 years [6]. Falls in the elderly population cause long-term immobilization and related complications. Therefore, balance disorders in elderly individuals are a symptom that leads to functional insufficiency. As a result of dynamic postural control, appropriate rehabilitation following the early detection of balance disorders and environmental modifications could prevent falls and increase an individual’s quality of life [7].

Balance disorders generate a significant healthcare burden due to the rise in hospitalization, morbidities, and mortalities in the elderly population [8]. Most of the patients who present to emergency services complain of balance disorders. In otology and neurology clinics, in which patients commonly present with balance disorders and dizziness, the rate of balance disorder is about 20% [9]. Thirteen percent of community-dwelling individuals aged 65–69 years and 46% of those aged ≥85 years have balance disorders [10]. There are many factors that lead to balance disorders, including cardiovascular diseases, metabolic diseases, musculoskeletal disorders, neurological disorders, visual and hearing disturbance, fear of falling, surgical operations, and specific medications [11].

Factors related to the risk of falling were taken into consideration in a report published by the World Health Organization (WHO) on the elderly in 2007. It was reported that balance disorders contribute to the occurrence of falls, and that balance exercises are a useful way to protect from falls. [12]. Goal number three of the Turkey Healthy Aging Implementation Program in the Healthy Aging Action Plan and Implementation Program published by the Ministry of Health for 2015–2020 includes a statement on the development of preventive and rehabilitative approaches to determine and decrease the risk factors leading to balance disorders, falling, and fear of falling in old age [13].

This study was conducted to determine the prevalence of balance disorders in the elderly population, identify the health (chronic illnesses, drug use) and social (leisure time activities) causes of balance disorders, and determine the role of balance disorders in falls.

## 2. Materials and Methods

### 2.1. Subjects

This was a cross-sectional study involving community-dwelling elderly individuals aged >65 years in a center in the Söke district of Aydın. A total of 607 elderly individuals with a mean age of 73.99 ± 6.6 (65–102) years were selected by the stratified random sampling method. The 65–74-year age group was stratified as the first group, the 75–84-year age group was the second group, and the ≥85-year age group was the third group. The study was performed as face-to-face surveys at the homes of elderly individuals. A total of 668 elderly individuals were asked to participate in the study, but those individuals who were bedridden, who were diagnosed with dementia, and who did not pass the Mini Mental State Examination (MMSE; cut-off score of 23) were not included in the study [14]. Eventually, 607 elderly individuals who agreed to voluntarily participate in the study formed the sample used for our study. Informed consent was given by all participants.

### 2.2. Evaluation Parameters

Elderly Identification Form: This form recorded information about the participants, including their age, gender, marital status, number of people in their household, economic level, education level, presence of illnesses, disability status, fall history, fear of falling, drugs used, presence of incontinence, nocturia, walking habits, leisure time activities, and body mass index. This form was created by the investigators.

Berg Balance Scale (BBS): This test, which was developed to measure balance performance in elderly individuals, consists of 14 instructions. Participants are given 0–4 points for each instruction according to their ability to perform the task; the maximum total score for the test is 56. It is a practical test that can be conducted in 15–20 min in the homes of the community-dwelling elderly individuals. The cut-off score for the test is 45. A score of 45–56 is an indication of an acceptable functional level. A score below 45 point is considered to indicate a balance disorder [15,16,17]. The Turkish translation and the transcultural adaptation of the BBS were previously studied in 60 elderly individuals with various comorbidities aged >65 years, and the validity and reliability have been reported [18].

### 2.3. Methods

The study started with the selection of Address-Based Population Registration System data with a stratified random sampling method for individuals over 65 years of age. The elders who were selected by this method represented elderly people living in the district center. Thus, the sample used was representative of the group of universe of elderly people in the district center. Four different teams were formed in the study, and these teams visited the elderly in their homes. The MMSE was applied to elderly people who participated voluntarily, who were not bedridden, and who had not been diagnosed with dementia. The Elderly Identification Form was given to participants to determine their socio-demographic and health characteristics, as well as to gather information about their free life activities. Sociodemographic, social, and health data were recorded on the form. Participants’ health reports were reviewed. The drugs they used were noted. The participants completed the BBS assessment, and their BBS scores were noted. The weights and heights of the participants were measured, and BMI values were calculated.

The present study was submitted to and approved by the Clinical Research Ethics Committee of the Ege University Faculty of Medicine (Decision Number: 16–3.2/7, Date: April 7, 2016). The study was conducted in accordance with the Declaration of Helsinki. All participants provided informed consent before being included in the study as a participant.

### 2.4. Statistics

Analysis of the data was done using SPSS (Version 18.0, SPSS Inc., Chicago, IL, USA). Chi-square tests were used to analyze balance disorders and fall variables. Univariate binary logistic regression analysis and multiple binary logistic regression analyses were conducted on all variables. For the results of the statistical analysis, *p*-values of <0.05 were considered significant.

## 3. Results

### 3.1. The Prevalence of Balance Disorders (Mean Value of BBS)

The BBS was used to measure the prevalence of balance disorders in the participants. In elderly individuals, the cut-off (sorter) value for balance disorders is 45 points [19,20]. The prevalence of a balance disorder in the elderly individuals in our study was 34.3%. The average BBS score was 43.49 ± 14.2.

### 3.2. The Relationship between Balance and Falls

For individuals with a balance disorder, 58.1% had fallen in the past year, and 90.3% had a fear of falling; in those without a balance disorder, 29.8% had fallen, and 60.3% had a fear of falling (Table 1).

### 3.3. Multiple Logistic Regression Analysis and the Effects of Sociodemographic, Medical, and Social Data on Balance Disorders

The sample of 607 participants was composed of 347 people aged 65–74 (first group), 214 aged 75–84 (second group), and 46 aged over 85 years old (third group). The association of age with balance disorders was statistically significant according to the multiple logistic regression analysis (*p* = 0.002). The prevalence of balance disorders increased as age increased. Balance disorders were 1.97 times higher in the second group (*p* = 0.006, OR = 1.97, 95% CI = 1.21–3.20) and 3.63 times higher in the third group (*p* = 0.003, OR = 3.63, 95% CI = 1.54–8.55), compared with the first group.

Three hundred and sixty-one of the 607 participants (59.47%) were females.

Participants included non-literate individuals, primary school quitters, primary school graduates (n = 276), secondary school graduates, high school graduates, and university graduates.

Participants included widows (divorced or had lost their husband/wife) and married individuals.

Participants included those living alone, with their spouses, with their spouses and children, only with their children, and with relatives/carers.

Considering economic situations, there were people without incomes, those receiving the elderly/widow/disabled wage, those receiving the retirement wage, and those using wages earned by their spouse.

The variables gender, education level, marital status, living status, and economic situation were not found to be statistically significant in our study, according to the multiple logistic regression analysis. 

Our study included elderly participants without any obstacles, elderly participants with a visual impairment, elderly participants with a hearing impairment, and elderly participants with disabilities. When the relationship between the disability status of elderly people and balance disorders was examined, with the group without obstacles taken as the reference group, participants with walking disabilities had a balance disorder 2.80 times higher than the reference group (*p* = 0.013, OR = 2.80, 95% CI = 1.24–6.33). The disability variable was found to be statistically significant (Table 2).

Two hundred and thirty-seven participants had fallen at least once in the past year. Fall history was statistically associated with the presence of a balance disorder in our study, according to the multiple logistic regression analysis. The prevalence of balance disorders among those who had fallen in the past year was 2.25 times higher than in those who had not fallen (*p* < 0.001, OR = 2.25, 95% CI = 1.46–3.46).

The prevalence of chronic disease was determined. Only 10.7% (n = 65) of the participants did not have any chronic illnesses. The number of chronic diseases was not statistically significantly associated with the prevalence of balance disorders in our study, according to multiple logistic regression analysis, but a significant association was shown in elderly people with four or more diseases (*p* = 0.047, OR = 3.54, 95% CI = 1.01–12.32).

Daily medication use was recorded for each participant. Hundreds of medications were classified according to their indications to determine which drugs can alter balance. For example, a participant using three medications (amlodipine [a selective calcium channel blocker], silazapril and hydrochlorothiazide [an angiotensin-converting enzyme inhibitor combination], and acetylsalicylic acid [an antithrombotic agent]) was categorized into a group of participants “using only one group of medications” (cardiovascular drug group). Medications were determined by group number instead of individual drugs. Our study included participants using neurological disease drugs, cardiovascular group drugs, diabetes medications, vertigo medications, thyroid medications, rheumatic disease drugs, pain killers, and depression group drugs. 12.8% of the participants (n = 78) did not use any medication (Table 2). The drug group use variable was not shown to be significantly associated with balance disorders in our study.

One hundred and seventy-seven of the participants stated that they could not hold urine during the day. In addition, 87 of the participants stated that they could not hold urine sometimes. Urinary incontinence was found to be statistically significantly associated with balance disorders (*p* = 0.002). Compared with the participants who had no difficulty with urinary incontinence, balance disorders were 2.4 times more prevalent in the participants who had urinary incontinence (*p* = 0.001, OR = 2.4, 95%CI = 1.46–3.95).

The study included elderly participants who did not urinate while sleeping at night and those who got up and went to urinate two or more times at night. Nocturia was not statistically significantly associated with balance disorders.

A total of 63.2% of the participants (n = 384) stated that they went out of the house and walked to go to the market, street market, mosque, coffee shop, or park. A lack of walking was statistically significantly associated with balance disorders. Balance disorders were 2.21 times more prevalent in the participants who did not walk than in those who walked to the market or those who took walks in the park (*p* = 0.001, OR = 2.21, 95% CI = 1.41–3.48).

Participants were asked about hobbies and interests to learn about leisure time activities. Participants reported eight types of hobby activities. These activities included reading activities; artistic activities, such as painting, music, and poetry; sports activities, such as swimming, fishing, and hunting; gardening and field work; making handcrafts; using the computer, foundation memberships; and mental games like chess and puzzles. Participation in leisure time activities was not statistically significantly associated with balance disorders, but, compared with the elderly who participated in three or more leisure time activities, the prevalence of balance disorders in participants who did not participate in activities was 3.4 times higher (*p* = 0.024, OR = 3.4, 95% CI = 1.17–9.88).

The BMI values of participants were measured. Individuals were classified as normal weight (<25), overweight (25–29.9), obese (30–34.9), or overly obese (≥35). BMI was not statistically significantly associated with balance disorders, but, compared with the participants with BMI < 25, balance disorders were approximately two times more common in obese (*p* = 0.044, OR = 2.09, 95% CI = 1.02–4.28) and overly obese participants (*p* = 0.019, OR = 2.53, 95% CI = 1.16–5.50).

## 4. Discussion

In this study, the prevalence of balance disorders was found to be 34.3% in the community-dwelling elderly. Older age, physical disability, the presence of incontinence, having a history of falls, having four or more chronic illnesses, not walking regularly, absence of free time activity, and obesity were found to be associated with an increased prevalence of balance disorders.

International studies have investigated balance disorders in elderly individuals living in the community. In a study conducted in the UK in 2008, the prevalence of balance disorders was found to be 21.5% among elderly individuals living in the community [21]. In a study conducted in the United States that was published in 2012 including participants aged ≥65 years with an average age of 74.4 years, the prevalence of balance disorders was approximately 20% [22]. In a study conducted in Scotland published in 1994, the prevalence of balance disorders was found to be 30% [23]. In another BBS-based study published in the US in 2006 including 101 community-dwelling volunteers aged >65 years, the prevalence of balance disorders was found to be 32% [24]. In our study, we found the prevalence of balance disorders to be 34.3%, which is in good agreement with the literature.

Şahin and colleagues performed a Turkish validity and reliability study of the BBS in 2008 including 60 healthy individuals aged >65 years. The average BBS score in that study was 47.63 ± 9.88 [18]. In a study conducted by Soyuer et al. using the BBS, the average BBS score was 45.42 ± 12.11 in nursing home residents [25]. In another study, the average BBS score was 41.3 ± 9 [26]. We found an average score of 43.49 ± 14.23 in our study.

In a study about the effect of age on balance disorders, a decrease in BBS scores with age was reported [27]. In another study, increased age was associated with decreasing BBS scores [28]. In our study, we also found that balance disorders were significantly more common in those aged 75–84 years and in those aged >85 years than in those aged 65–74 years.

In a study published in 2012, in which the effect of gender on the prevalence of balance disorders was examined, the prevalence of balance disorders was reported to be higher in females than in males [22]. In another study published in 2013, BBS scores were lower for female participants [27]. However, gender was not found to be a meaningful variable in our study.

Regarding the relationship between visual disturbance, balance, and falls, visual disturbance was found to be associated with the prevalence of falling [29]. In another study, elderly patients with visual disorders were found to have lower balance scores than a control group [30]. In another study, peripheral visual loss was reported to have a negative effect on balance control [31]. In our study, although overall disability status was significantly associated with balance disorders, visual disability had no effect on balance. At the same time, walking impairment was associated with balance.

In a study published in 2012, one-third of participants with balance disorders participated in no exercise-related activities or social activities [22]. In our study, participants who did not walk and who did not participate in any free time activities had a high likelihood of having a balance disorder. Our data match those reported by others. In a study published in 2019, the effect of exercise on falls was reported. Sherrington et al. found that participation in exercise mainly involving balance and functional exercises, plus resistance exercise, was associated with a reduction in falls. However, Sherrington et al. did not find enough evidence to determine the effect of walking programs on falls [32]. In contrast, in our study, we found a lower prevalence of balance disorders in those who walked regularly than in those who did not walk.

We think that the causes of balance disorders in elderly people with a walking disability and those who do not walk regularly may be sarcopenia and demineralization. It is known that the most important muscle for balance is the quadriceps femoris. We think that the loss of this muscle has a negative effect on balance in elderly people who do not walk regularly. In addition, loss of minerals from bone and decreased signal frequency from the proprioceptive receptors may have negative effects. As a result, we advise elderly people to walk regularly.

When we look at the relationship between postural balance and falls, posturographic vestibular rehabilitation has been reported to improve balance in elderly individuals and to reduce the number of falls [33]. Impaired balance has been reported as one of the long term risk factors for falls in men [34]. In another study, participants were classified into the non-fall group, one-time fall group, and repeated fall group. Both the dynamic balance and static balance scores were found to be higher in the non-fall group than in the one-time fall group and repeated fall group [35]. In our study, the prevalence of balance disorders was found to be significantly higher in people with a fall history.

In a study that was conducted to show the relationship between balance disorders and a fear of falling, the presence of balance, gait, and cognitive disorders was reported to be significantly higher in elderly individuals with a fear of falling [36]. In another study that used the Berg Balance Scale, lower BBS scores were found in the group with a fear of falling compared to those in the group without a fear of falling [27]. In our study, the presence of balance disorders was found to be significantly higher in those who had a fear of falling than in those who did not have a fear of falling (*p* < 0.001) (Table 1)

In a study on obesity in the elderly, an increased Body Mass Index (BMI) was associated with decreases in both dynamic stability and balance in the elderly [37]. In a study published in 2018, older participants were classified into obese, normal weight, and weak groups according to Body Mass Index scores. The one-leg standing time test was applied to determine balance scores. In the obese group, the one-leg standing time was much shorter than in the normal weight group of community-dwelling elderly women [38]. In our study, the prevalence of balance disorders was found to be higher in obese (BMI = 30–34.9) and overly obese (BMI ≥ 35) elderly individuals (*p* = 0.044 and *p* = 0.019, respectively) than in individuals of normal weight.

Balance disorders, incontinence, fall history, and age have been associated with each other in many studies published in this area [39,40]. These variables were found to be significant in the multiple logistic regression analysis in our study, too.

Our study had both limitations and significant contributions. In terms of limitations, some of the survey data were determined from the statements of elderly individuals—fall history, for example. In addition, the participants in our study consisted of elderly people living in a district center, and different socio-demographic data could be obtained from elderly people living in other parts of the country. In terms of contributions, to our knowledge, no previous study in this area has been conducted in Turkey using the stratified random sampling method to interview a large sample of community-dwelling elderly individuals face-to-face at their homes. We were able to get the right data because we applied the BBS test. In addition to academic data, our study also guided a local government social project. Walking support materials (walking stick, walker, wheelchair) were given to the elderly who were identified as having a balance disorder. Our work has created awareness in the community (see Appendix A).

Our research data may be used as reference for other academic studies in this area. The identification of the relationship between equilibrium and falling in the elderly and the emergence of many health and social causes of balance disorders are very valuable. More work is needed to determine the mechanisms behind each of these reasons. Awareness of these causes is needed for both individual and social preventive health practices. Following the collection of these data, people have learned that they need to change their lifestyles to protect their health. The local government has launched projects to support elderly people after obtaining these data. It has opened courses where seniors can spend their leisure time, with picture, music, craft, and folklore courses. Using the sociodemographic data of the elderly collected in this study, a journal called “Old Age Atlas of Söke” was published in the district. Söke Municipality published this journal to district people. This journal increased the awareness of people in the district about old age and old age problems. The social contributions of the study were appreciated by people in the district in addition to the academic contributions.

Balance disorders, which are considered to have several biological and social etiologies, are a major geriatric problem which lead leading to falling and increased morbidity and mortality rates. The development of national health and social policies that address the underlying causes of this problem and the introduction of preventive health care services should be the primary steps towards helping today’s increasingly elderly population. The concept of “age-friendly” should be widespread in all segments of the society, including the private sector and public services.

## Figures and Tables

**Table 1 geriatrics-04-00037-t001:** The relationship between balance and falling.

	Fell Last Year			Fear of Falling		
	Yes (n = 237)	No (n = 370)	Chi-square (*p*-value)	Yes (n = 427)	No (n = 178)	Chi-square (*p*-value)
Balance disorder (n = 208)	121 (58.1%)	87 (41.8%)	X^2^ = (*p* < 0.001)	187 (90.3%)	20 (9.7%)	X^2^ = (*p* < 0.001)
No balance disorder (n = 389)	116 (29.8%)	283 (72.7%)		240 (60.3%)	158 (39.7%)	

Chi-square test, *p* ˂ 0.05: statistically significant.

**Table 2 geriatrics-04-00037-t002:** Univariate logistic regression analysis and multiple logistic regression analysis.

Variant	Reference Group (n)	Other Groups (n)	Univariate Logistic Regression			** Multiple Logistic Regression		
			P	OR	95% C.I.	P	OR	95% C.I.
Age						0.002 *		
	6574 years (n = 347)	7584 (n = 214)	<0.001	2.34	1.623.36	0.006 *	1.97	1.213.20
		≥85 (n = 46)	<0.001	4.24	2.258.01	0.003 *	3.63	1.548.55
Gender	male (n = 246)	female (n = 361)	<0.001	2.30	1.603.30	0.211	0.65	0.34–1.26
Education						0.162		
	university-high (n = 54)	non-literate (n = 160)	<0.001	8.20	3.3220.24	0.071	2.72	0.918.08
		quit prim. s. (n = 90)	0.001	5.09	1.9713.14	0.082	2.69	0.888.20
		primary s. (n = 276)	0.011	3.15	1.297.66	0.259	1.80	0.645.00
		secondary s. (n = 27)	0.044	3.36	1.0311.01	0.061	3.85	0.9315.84
Marital status	married (n = 407)	widow (n = 200)	<0.001	2.95	2.074.21	0.494	1.65	0.397.04
Living						0.946		
	with spouse (n = 291)	living alone (n = 115)	<0.001	3.20	2.035.03	0.466	1.76	0.388.06
		spouse-child (n = 113)	0.359	1.25	0.772.03	0.584	1.18	0.642.15
		children (n = 75)	<0.001	3.29	1.945.57	0.582	1.52	0.336.92
		relative/carer (n = 13)	0.273	1.90	0.605.99	0.665	1.48	0.248.93
Economic						0.123		
	retired (n = 298)	no income (n = 140)	0.003	1.91	1.242.94	0.105	1.82	0.883.78
		aged wage (n = 53)	<0.001	3.51	1.926.41	0.544	1.29	0.562.95
		wage spouse(n = 116)	<0.001	2.83	1.804.44	0.397	0.72	0.351.51
Disability						0.027 *		
	no (n = 463)	blind (n = 42)	0.005	2.48	1.314.69	0.082	2.03	0.914.51
		hearing imp. (n = 57)	0.126	1.56	0.882.75	0.776	0.89	0.421.90
		walking imp. (n = 45)	<0.001	6.10	3.1011.99	0.013 *	2.80	1.246.33
Fall history	no (n = 370)	yes (n = 237)	<0.001	3.39	2.394.81	˂0.001 *	2.25	1.463.46
Number CD						0.205		
	no (n = 65)	1 (n = 117)	0.326	1.52	0.653.52	0.390	1.70	0.505.74
		2 (n = 160)	0.004	3.17	1.456.89	0.152	2.43	0.728.24
		3 (n = 109)	0.001	3.90	1.748.70	0.090	2.98	0.8410.55
		≥4 (n = 156)	˂0.001	6.55	3.0314.15	0.047 *	3.54	1.0112.32
Number MG						0.831		
	no (n = 78)	1 (n = 160)	0.265	1.47	0.742.91	0.450	0.65	0.221.95
		2 (n = 164)	0.011	2.37	1.224.59	0.524	0.70	0.232.09
		≥3 (n = 205)	<0.001	4.27	2.258.09	0.697	0.80	0.262.44
Incontinence						0.002 *		
	no (n = 343)	yes (n = 177)	<0.001	5.30	3.567.89	0.001 *	2.40	1.463.95
		sometimes (n = 87)	<0.001	2.75	1.664.54	0.057	1.81	0.983.35
Nocturia	no (n = 280)	yes (n = 327)	<0.001	2.83	1.984.05	0.577	1.14	0.711.81
Walking	yes (n = 384)	no (n = 223)	<0.001	4.18	2.935.97	0.001 *	2.21	1.413.48
LTA						0.079		
	3+ (n = 77)	no (n = 60)	<0.001	9.88	4.1723.40	0.024 *	3.40	1.179.88
		1+2 (n = 470)	<0.001	4.08	1.988.40	0.097	2.04	0.874.77
BMI						0.093		
	<25 (n = 114)	2529.9 (n = 217)	0.340	1.29	0.762.19	0.206	1.54	0.783.03
		3034.9 (n = 170)	0.002	2.36	1.394.04	0.044 *	2.09	1.024.28
		≥35 (n = 106)	<0.001	3.25	1.825.82	0.019 *	2.53	1.165.50

* Statistically significant (*p* < 0.05), OR: Odds ratio, C.I.: Confidence interval, CD: Chronic disease, MG: Medication group, LTA: Leisure time activities, BMI: Body Mass Index, prim. s.: Primary school, s.: School, imp.: Impairment, ** Multiple logistic regression analysis with enter method.

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
