# Peer review of "The Relationship of Balance Disorders with Falling, the Effect of Health Problems, and Social Life on Postural Balance in the Elderly Living in a District in Turkey"

_geriatrics, 2019, doi:10.3390/geriatrics4020037_

Round 1

Reviewer 1 Report

1.     The study has a very good intention and data collection was done correctly, but statistical analyses need to be improved. Since the study has a large sample size, almost all univariate analyses became significant. (Senses data do not conduct statistical analysis, beucase all analyses become significant with a large sample size.) For logistic regression, instead of conducing it for each variable, use blocks for covariates with forward methods. Multicollinearity should be examined prior to the analysis. Age, marital status, and living situation may be related and economic level and marital status may be related.  In this way, we will understand what are important factors that have storing relationship with balance disorder.  

2.     In discussion, write limitations and contribution of the study.    

Author Response

Dear reiewer, thanks for your comments about our manuscript

We added a more descriptive paragraph in the methods section. We only used Chi-square test for fall and fear of falling. We re-presented all the results more clearly through a table and LR analysis by subtracting T test and Chi square tests and tables.We applied Multiple Logistic Regression (MLR) analysis, thus we used blocks between variables on MLR, eventually we found the effect of the eight variables on balance disorder. We wrote limitations and contribution of the study in discussion.Kind regardsAuthors

Reviewer 2 Report

General Comments:

This is an important area of research. However, there are major deficiencies in the manuscript that are of concern.  While I think it is important to understand the prevalence of selected health characteristics within a selected population, the manner in which these data are presented lack interpretation and context that would highlight the unique contributions of this work.  This manuscript reads more like a technical report than a scholarly endeavor. The approach is off putting because there is a lengthy list of “evaluation parameters”; 19 categories of data many with up to 10 levels of comparators.  The investigators simply ran a series of T-test and described the results item by item like a laundry list.  This was repeated almost exactly in the discussion.  A line by line repeat of the results, only this time, for each result, there were one or two citations referencing similar findings in other studies.  This only highlighted the lack of innovation of this study.  There was no attempt for an interpretation of the meaning of the findings.  Would recommend a consultation with a statistician to determine if a different approach might be used or whether there should be an adjustment for multiple comparisons or some higher level of modeling with appropriate adjustment for the various groupings of data. It is also not clear from the analysis description whether the data runs were simply one-on-one comparisons by T-Test, chi square and the other methods employed, or whether the outcomes were compared against a reference group.  For example, in the comparisons of education with balance disorders, Table 3 shows 2 comparisons non literate and primary schools and shows a p-value for the comparison between these two groups.  But then in the text there are 5 categories of educational status (Primary, secondary, etc) and it is not clear what the associations with these variables are.  Was primary school declare the reference group and the other comparisons were with primary school and only non-literate was significant?  This approach is throughout and should be rectified after appropriate statistical consideration.  The authors should think carefully about their data and try to reframe the manuscript in a thoughtful manner that points out potential areas where this work represents a unique and important contribution to the literature.

Author Response

Dear reiewer, thanks for your comments about our manuscript We added a more descriptive paragraph in the methods section We removed the tables such a laundry list, T test and chi square test results. We re-presented all the results more clearly through a table and LR analysis. We avoided repeating the same sentences in the results and discussion. We consulted with a statistician, we used Multiple logistic regression analysis and found factors effective on balance disorder elderly, added additional literature.  We could not send for edit because we didn't have time when we finished the revision, We want sent our manuscript to MDPI English Editing service, while under concideration. Kind regards Authors

Round 2

Reviewer 1 Report

For Authors

This manuscript has improved a lot, but there is room to improve.

See attachment.

Author Response

Dear reviewer,

First of all, thank you very much for guiding us with your valuable comments.

let's look at your list of advice respectely

1.       In Abstract, Line 28 - 31, we changed and rewritten the conclusions sentences

2.       In Subject, Line 80, we wrote cut-off score of MMSE and added the reference

3.       In Results, we removed Table 1

4.       We arranged Table 2 like you did

5.       We cosulted with our statistician. Results of univariate logistic regression are not meaningful compared to values of odds ratio in our study. You are right. Thats why, we used multiple logistic regrssion analysis with enter method. We  wrote variables that show significant p-values in our study.

6.       We had used method of forward stepwise (likelihood ratio) before. But, we changed method according to your valuable comments. We entered all variables and used enter method. We obtained more correct results, thank you very much.

7.       We wrote prevalence of balance disorder and significant results for our study at the beginning of the discussion.

8.       Our manuscript was edited by MDPI English Editing Service

Kind regards,

Dr. Değer (corresponding author)

Reviewer 2 Report

The authors have put forth a good effort in addressing prior feedback. The statistics are more clearly defined. The discussion still reads like a laundry list with little interpretation of findings beyond increasing the community awareness of the problems. On the positive this data contributed towards a government initiative.

Would consider revising the title to include that the sample is from Turkey. Since this is one of the innovations sited by the authors.

In the abstract the following sentence should be revised "Older age, 21 marital status (widow), physical disability, increased numbers of chronic illnesses, the presence of incontinence, history of falls, do not walk regularly and limited free time activities were factors which increased the prevalence of balance disorder" it should be revise to state that the factors were "associated with increased prevalence of balance disorder" instead of saying that they were "factors which increased the prevalence" since this is a cross sectional study.

Author Response

Dear reviewer

First of all, thank you very much for guiding us with your valuable comments.

let's look at your list of advice respectely

1.       In Discussion, Line 331-337, we wrote interpretation about walking disability and walk regularly. Line 379-393, we added a paragraph about the benefits of the results of the study to the community. We changed statisticaly method, and we used enter method in multiple logistic regression analysis. We revised the discussion.

2.       We added "living in a district in Turkey" to the title.

3.       In Abstract, we made changes as you specified

4.       Our manuscript was edited by MDPI English Editing Service

Kind regards,

Dr. Değer (corresponding author)